# Deep Learning-Based Glioma Segmentation of 2D Intraoperative Ultrasound Images: A Multicenter Study Using the Brain Tumor Intraoperative Ultrasound Database (BraTioUS)

**DOI:** 10.3390/cancers17020315

**Published:** 2025-01-19

**Authors:** Santiago Cepeda, Olga Esteban-Sinovas, Vikas Singh, Prakash Shetty, Aliasgar Moiyadi, Luke Dixon, Alistair Weld, Giulio Anichini, Stamatia Giannarou, Sophie Camp, Ilyess Zemmoura, Giuseppe Roberto Giammalva, Massimiliano Del Bene, Arianna Barbotti, Francesco DiMeco, Timothy Richard West, Brian Vala Nahed, Roberto Romero, Ignacio Arrese, Roberto Hornero, Rosario Sarabia

**Affiliations:** 1Department of Neurosurgery, Río Hortega University Hospital, 47014 Valladolid, Spain; oestebans@saludcastillayleon.es (O.E.-S.); iarrese14@yahoo.es (I.A.); rsarabia@saludcastillayleon.es (R.S.); 2Department of Neurosurgery, Tata Memorial Centre, Homi Bhabha National Institute, Mumbai 400012, Maharashtra, India; drvikaskumarsingh@gmail.com (V.S.); drpmshetty@gmail.com (P.S.); aliasgar.moiyadi@gmail.com (A.M.); 3Department of Imaging, Charing Cross Hospital, Fulham Palace Rd, London W6 8RF, UK; luke.dixon03@imperial.ac.uk; 4Hamlyn Centre, Imperial College London, Exhibition Rd, London SW7 2AZ, UK; a.weld20@imperial.ac.uk (A.W.); stamatia.giannarou@imperial.ac.uk (S.G.); 5Department of Neurosurgery, Charing Cross Hospital, Fulham Palace Rd, London W6 8RF, UK; g.anichini@imperial.ac.uk (G.A.); s.camp@imperial.ac.uk (S.C.); 6UMR 1253, iBrain, Université de Tours, Inserm, 37000 Tours, France; ilyess.zemmoura@univ-tours.fr; 7Department of Neurosurgery, CHRU de Tours, 37000 Tours, France; 8Neurosurgery Department, ARNAS Civico Di Cristina Benfratelli Hospital, 90127 Palermo, Italy; robertogiammalva@live.it; 9Department of Neurosurgery, Fondazione IRCCS Istituto Neurologico Carlo Besta, Via Celoria 11, 20133 Milan, Italy; macs.delbene@gmail.com (M.D.B.); ariannabarbotti.ab@gmail.com (A.B.); francesco.dimeco@istituto-besta.it (F.D.); 10Department of Pharmacological and Biomolecular Sciences, University of Milan, 20122 Milan, Italy; 11Department of Oncology and Hematology-Oncology, Università Degli Studi di Milano, 20122 Milan, Italy; 12Department of Neurological Surgery, Johns Hopkins Medical School, Baltimore, MD 21205, USA; 13Department of Neurosurgery, Massachusetts General Hospital, Mass General Brigham, Harvard Medical School, Boston, MA 02114, USA; trwest@mgh.harvard.edu (T.R.W.); bnahed@mgh.harvard.edu (B.V.N.); 14Biomedical Engineering Group, Universidad de Valladolid, 47011 Valladolid, Spain; rober8ro@gmail.com (R.R.); roberto.hornero@uva.es (R.H.); 15Center for Biomedical Research in Network of Bioengineering, Biomaterials and Nanomedicine (CIBER-BBN), 47011 Valladolid, Spain; 16Institute for Research in Mathematics (IMUVA), University of Valladolid, 47011 Valladolid, Spain

**Keywords:** ultrasound, segmentation, deep learning, CNN, Glioma, brain tumor

## Abstract

This study explores the use of artificial intelligence to improve the accuracy of intraoperative ultrasound (ioUS) imaging for glioma segmentation during neurosurgery. By training a deep learning model on data from multiple centers, this research demonstrates the potential for automated tumor delineation, despite challenges such as image noise and variability. The model was tested on independent datasets and showed strong performance overall, although external validation highlighted areas for improvement. Notably, the model demonstrated generalizability across ioUS systems from different scanner types and manufacturers, underscoring its robustness in diverse clinical settings. These findings emphasize the feasibility of using AI to enhance ioUS imaging, paving the way for more precise and efficient tumor resections in clinical practice.

## 1. Introduction

Intraoperative ultrasound (ioUS) has been an essential tool in neurosurgery for several decades because of its portability, cost-effectiveness, and flexibility within the surgical workflow [1,2,3,4]. These characteristics make ioUS a vital resource for real-time imaging that can inform critical intraoperative decisions. Nevertheless, ioUS has recognized limitations: image quality is often affected by noise and artifacts, oblique acquisition angles can obscure anatomical boundaries, and interpretation can be challenging, particularly for surgeons with limited experience in ultrasound imaging. These limitations highlight the need for solutions that improve ioUS interpretability while maintaining its practical and cost-effective characteristics. Solutions such as tracked ioUS probes and pre-operative MRI fusion have been proposed to assist image interpretability [1,5,6]. Navigated ioUS is, however, expensive and thus not universally available. While useful for orientation and localization, reference MRI is not contemporaneous, and there remains high operator dependence and cognitive burden on the interpretation of the live ioUS image itself. Moreover, reliance on complex and costly technologies contradicts the core advantages of ioUS—its simplicity and affordability.

Tumor segmentation, the process of delineating tumor boundaries in medical images, is a complex task that has spurred extensive research, competition, and collaborative initiatives, particularly in MRI-based studies of brain tumors [7,8]. In contrast, research into ioUS segmentation has only recently gained momentum, driven partly by advancements in ultrasound image quality over the past decade and the availability of public datasets supporting research in this area [9,10,11,12]. Precise tumor segmentation via ioUS could significantly benefit neurosurgery by providing clearer delineation of resection margins and aiding interpretation for surgeons who are less familiar with ioUS.

Previous studies have explored the application of convolutional neural networks (CNNs) to train models for tumor and surgical cavity segmentation in neuro-oncology [13,14,15,16,17,18,19], often relying on public datasets [9,10,11,12]. While these datasets are invaluable, they are frequently limited by variability in image quality, tumor types, and acquisition modalities. For example, Canalini et al. [14] focused on the segmentation of anatomical structures such as sulci and falx, which, while valuable, do not directly address tumor delineation. Carton et al. [20] and Zeineldin et al. [18] concentrated solely on surgical cavity segmentation, which provides limited assistance in glioma boundary detection. Dorent et al. [13] integrated segmentations derived from pre-operative MRI, reducing their applicability in purely ioUS-based workflows. Angel Raya et al. [17] expanded segmentation to include metastases, diverging from a glioma-specific focus. Faanes et al. [19] combined annotated ioUS images with coregistered MRI-derived segmentations, achieving promising results that highlight the potential of multimodal imaging, albeit with additional complexity and reliance on multiple imaging modalities. Carton et al., while focusing on low-grade gliomas, worked with a small patient cohort, restricting the generalizability of their findings.

Moreover, while these studies have made important strides, many rely on public datasets with constrained scopes and limited diversity. A common challenge is the absence of validation in external cohorts, which limits the reproducibility and robustness of the proposed methods in broader clinical applications.

In this study, we aimed to address the challenge of accurate and automated glioma segmentation in ioUS images by developing a CNN model trained on a multicenter dataset provided by the Brain Tumor Intraoperative Ultrasound Database (BraTioUS). This dataset includes images acquired using various scanners and diverse acquisition protocols, enabling the model to handle the inherent variability of ioUS data. Our work focuses on both high- and low-grade gliomas, with the aim of evaluating the model’s performance on external datasets to ensure its reproducibility and robustness. This approach seeks to overcome key limitations in existing methodologies, advancing the interpretability and clinical applicability of ioUS for glioma surgery.

## 2. Materials and Methods

### 2.1. Dataset

The primary dataset for this study is derived from the BraTioUS Consortium (ClinicalTrials.gov Identifier: NCT05062772). This consortium includes contributions from Río Hortega University Hospital, Valladolid, Spain (RHUH); Tata Memorial Center, Mumbai, India (TMC); Istituto Neurologico “Carlo Besta,” Milan, Italy (INCC); University of Palermo, Italy (UPALER); Le Centre Hospitalier Régional Universitaire de Tours, France (CHRUT); and Massachusetts General Hospital, Boston, MA, USA (MGH). The dataset comprises intraoperative ultrasound (ioUS) images from patients who underwent brain tumor surgery between 2018 and 2023.

For this study, we included only patients diagnosed with glioma, according to the 2021 WHO classification of CNS tumors [20]. We selected only pre-resection B-mode images, excluding cases with other histopathological diagnoses, images of suboptimal quality, or artifacts that impeded processing and interpretation.

A second data source, the publicly available ReMIND dataset [9], was included and subjected to the same selection criteria. The combined dataset was randomly split into training and testing cohorts, stratified by institution of origin, at a 70/30 ratio. Two independent external validation cohorts, the Imperial College NHS Trust London cohort [21] and the RESECT-SEG dataset [11], were used to assess the model’s performance and generalizability. The use of anonymized data was approved by the Research Ethics Committee (CEIm) at Río Hortega University Hospital, Valladolid, Spain (approval number 21-PI085). A schematic representation of the workflow followed in this study is provided in Figure 1, outlining the key steps and processes implemented throughout the methodology.

### 2.2. Preprocessing and Ground Truth Segmentation

For each patient, one 2D slice displaying the largest tumor diameter was selected. In cases with 3D ioUS volumes, which are distinct from native 2D acquisition volumes, they were decomposed into individual 2D axial slices, from which the most relevant slice was subsequently chosen. The tumors in each selected 2D slice were manually segmented using ITK-SNAP software (Version 4.0.1, http://itksnap.org, accessed on 1 November 2024), excluding large necrotic or cystic regions from the segmentation. Image selection and segmentation were performed by a neurosurgeon with 12 years of experience in medical imaging, particularly in the interpretation and analysis of intraoperative ultrasound. Figure 2 shows several examples of ground truth segmentation across the datasets used in this study.

### 2.3. nnU-Net Framework

We utilized the nnU-Net framework [22,23] in its 2D configuration, a self-configuring deep learning pipeline specifically designed for medical image segmentation. Our implementation uses the default hyperparameters provided by nnU-Net. The model was trained from scratch over 1000 epochs using five-fold cross-validation. The loss function combined Dice and cross-entropy components to optimize segmentation accuracy by balancing pixelwise classification with spatial overlap. Soft Dice loss is defined as$$\mathrm{Soft}\;\mathrm{Dice}\;\mathrm{Loss}\;=1-\frac{2\sum_{i=1}^{N}\;p_{i}g_{i}}{\sum_{i=1}^{N}\;p_{i}+\sum_{i=1}^{N}\;g_{i}}$$
where N represents the total number of pixels, $p_{i}$ notes the predicted probability for the i-th pixel, and $g_{i}$ is the ground truth label for the i-th pixel. Cross-entropy loss is expressed as$$\mathrm{Cross}\text{-}\mathrm{Entropy}\;\mathrm{Loss}\;=-\frac{1}{N}\sum_{i=1}^{N}\;\left[g_{i}log\left(p_{i}\right)+\left(1-g_{i}\right)log\left(1-p_{i}\right)\right]$$

In our training process, the total loss function was the sum of these two components:Total Loss = Soft Dice Loss + Cross-Entropy Loss

We employed data augmentation techniques such as rotation, scaling, Gaussian noise, blur, brightness and contrast adjustments, low-resolution simulations, gamma correction, and mirroring to increase model robustness. No postprocessing was applied to the predicted segmentations, maintaining the raw output from the model.

### 2.4. Evaluation Metrics

To assess model performance on the hold-out testing set and the external validation set, we used the USE-Evaluator [24], a tool that provides a more comprehensive evaluation than conventional metrics do, particularly for clinical datasets with complexities such as small residual tumor labels or cases with empty annotations due to complete resection. The USE-Evaluator includes overlap metrics, such as the Dice similarity coefficient (DSC) and intersection over union (IoU) and distance metrics, such as the 95th percentile Hausdorff distance (HD 95) and average symmetric surface distance (ASSD). This range of metrics allows for a nuanced evaluation of segmentation performance, capturing both spatial accuracy and boundary delineation precision.

### 2.5. Computational Resources

Both training and evaluation were conducted on a machine equipped with an Intel Core i9 processor, 64 GB of RAM, and an NVIDIA RTX 3090 GPU with 24 GB of memory. The nnU-Net model was trained via Python 3.9 and PyTorch version 2.1.1 with CUDA 12.1 support.

## 3. Results

The BraTioUS dataset includes a total of 154 patients, 152 of whom met the selection criteria for this study. From the ReMIND dataset, 45 patients were selected from an initial pool of 114. For training, a total of 141 images—one per subject—distributed across the following institutions were included: RHUH (41 subjects), ReMIND (32 subjects), TMC (25 subjects), CHRUT (21 subjects), UPALER (11 subjects), INCC (7 subjects), and MGH (4 subjects).

The hold-out testing cohort comprised 56 subjects, with one image per subject, distributed as follows: RHUH (17 subjects), ReMIND (13 subjects), TMC (10 subjects), CHRUT (8 subjects), INCC (3 subjects), UPALER (4 subjects), and MGH (1 subject). An additional external validation cohort, RESECT-SEG, consisted of 23 subjects, whereas the Imperial NHS cohort included 30 subjects. Comprehensive details regarding imaging acquisition protocols and patient characteristics are provided in the associated publications [11,21].

For the external validation cohorts, 3,501 valid segmentations were obtained from a dataset of 7,507 2D images generated by axial slicing of 3D volumes in the RESECT-SEG cohort. In contrast, the Imperial-NHS cohort, comprising 30 subjects, utilized a single native 2D image per subject.

In the training cohort, the mean age ranged from 42.49 ± 15.16 to 67.33 ± 12.98 years, with a balanced distribution of sexes across centers. Most of the tumors were WHO grade 4 (74.6%), with smaller proportions of grades 2 (9.6%) and 3 (10.2%). IDH status was predominantly wild-type (59.9%), followed by mutant cases (22.3%), while 16.8% lacked IDH data. In the dataset, 64.9% of the ultrasound acquisitions were performed in 2D mode, whereas 35% were performed in 3D mode. The probe types varied between curved and linear, and the frequencies ranged from 3–15 MHz. Table 1 details the acquisition parameters, including scanner manufacturers and imaging protocols across different centers.

The performance evaluation across datasets and centers demonstrated variability in segmentation metrics. In the hold-out testing cohort (56 patients), the model achieved a median DSC of 0.90, ASSD of 8.51 mm, HD95 of 29.08 mm, IoU of 0.82, precision of 0.91, and sensitivity of 0.91. Individual centers showed variation, with RHUH and CHRUT presenting higher HD95 values (38.36 mm and 60.72 mm, respectively) and INCC exhibiting the lowest performance (DSC: 0.76, IoU: 0.61). For external validation on the RESECT-SEG cohort (23 patients), performance was lower, with a median DSC of 0.65, ASSD of 14.14 mm, HD95 of 44.02 mm, IoU of 0.48, precision of 0.84, and sensitivity of 0.61. In contrast, for the Imperial-NHS cohort (30 subjects), the model achieved a median DSC of 0.93, ASSD of 8.58 mm, HD95 of 28.81 mm, IoU of 0.86, precision of 0.94, and sensitivity of 0.91. Figure 3 shows the performance metrics by center in the form of boxplots, while Figure 4 presents examples of predictions across the different datasets. Table 2 summarizes the model’s performance on the testing and external validation datasets.

## 4. Discussion

In this study, we developed a glioma segmentation model for 2D intraoperative ultrasound (ioUS) images using a multicenter cohort, achieving promising reproducibility and generalizability. Our work’s strengths lie in the careful selection of high-quality images and rigorous tumor segmentation, which ensures consistency across data sources. To our knowledge, this is the first model focused exclusively on glioma segmentation, trained on a multicenter dataset integrated with public data sources.

Despite the lower performance observed in the RESECT-SEG external validation cohort compared to both the hold-out testing cohort and the Imperial-NHS cohort, it is critical to contextualize these results. The RESECT-SEG dataset included all axial 2D slices derived from 3D volumes of 23 patients, introducing substantial variability in the size and complexity of the ground truth segmentations. Compared to the 2D US images, the 3D volumes acquired on an earlier generation system are also of much lower spatial and temporal resolution and suffer from greater noise, more artifacts, and the potential for reconstruction errors. Furthermore, the cohort predominantly consisted of patients with low-grade gliomas, which can have more diffuse, infiltrative boundaries that are more challenging to delineate compared to higher-grade tumors. These factors contribute to the inherent difficulty of segmenting such cases, even for clinical experts [21].

In contrast, the Imperial-NHS cohort, which utilized high-quality native 2D images from 30 subjects, yielded significantly better performance metrics, comparable to those observed in the hold-out testing cohort. This not only underscores the superior quality of imaging data and segmentations in the Imperial-NHS cohort but also highlights the generalizability of our model across diverse external datasets. Together, these findings demonstrate the model’s robustness and adaptability while emphasizing the critical role of image quality and annotation consistency in achieving optimal segmentation performance.

Medical image segmentation remains a substantial challenge in the application of artificial intelligence. Owing to their inherent characteristics, ultrasound images pose unique difficulties. Recent improvements in ultrasound quality have increased the signal-to-noise ratio and contrast-to-noise ratio, advancing the potential of ultrasound in clinical applications. Beyond anomaly detection, ultrasound has proven useful in procedural guidance, tissue characterization, and the exploration of biological tissue correlations [25].

However, boundary delineation in ultrasound is limited by factors such as acoustic impedance differences, insonation angles, signal attenuation, and speckle artifacts, which contribute to the characteristic granular appearance of ultrasound. While speckle is often treated as noise, it can also reflect tissue heterogeneity, as its local brightness pattern varies with tissue structure [25,26]. Although beneficial in some applications, these unique properties make tumor boundary detection in ioUS particularly challenging.

Several studies have applied CNNs to segment various structures in ultrasound, including applications in echocardiography, breast ultrasound, and gynecology [27,28,29,30,31,32,33]. However, few studies have focused on brain tumors. For example, Canalini et al. [14] trained a CNN for 3D segmentation of hyperechogenic structures such as sulci and falx cerebri, which serve as landmarks to improve alignment during glioma resection. Similarly, Carton et al. [15] addressed brain shift challenges by segmenting resection cavities in ioUS images via U-Net models, achieving a mean Dice score of 0.72 with a 3D network, and demonstrated rapid performance with the 2D model. They also trained multiclass models to delineate multiple structures, leveraging interclass spatial relationships to improve performance [16].

Angel-Raya et al. [17] evaluated different methods to segment brain tumors in 3D ioUS, achieving the highest accuracy with a semiautomatic approach, whereas Zeineldin et al. [18] reported that a transformer-based TransUNet outperformed a standard UNet for resection cavity segmentation, with an average Dice score of 93.7. Additionally, Dorent et al. [13] proposed a framework that uses synthetic ioUS images generated from pre-operative MR images to train a patient-specific model in real time, enhancing tumor delineation without complex tracking systems. Recently, Faanes et al. [19] showed that pre-operative MRI annotations can substitute manual iUS annotations for training 2D tumor segmentation models, with comparable performance and improved results when both modalities are combined, highlighting the value of integrating MRI and ultrasound for robust segmentation.

The key distinguishing feature of our study, compared to the above-mentioned publications, is the unique composition of the dataset employed. Most previous studies rely on RESECT [10,11] and BITE [12], which contain only 3D acquisitions that often require conversion to 2D slices. This conversion process can limit segmentation quality, since many slices lack tumor information, and 3D image resolution is often inferior to native 2D images. Furthermore, these datasets predominantly include low-grade gliomas. Our study combines data from multiple centers and includes the ReMIND dataset, carefully selecting cases with histologically confirmed gliomas and adequate image quality for manual segmentation.

Selecting the 2D slice with the largest tumor diameter was essential to maximize tumor representation and segmentation quality, as including slices with limited tumor information could introduce noise. High- and low-grade gliomas differ significantly in imaging characteristics; high-grade gliomas tend to have clearer boundaries, whereas low-grade gliomas appear more diffuse, complicating boundary delineation. Our dataset balanced these variations across histological types, institutions, and imaging sources, aiming to enhance the model’s reproducibility and generalizability.

This study’s limitations include a relatively small sample size and potential observer bias from the manual selection of the largest tumor slice and the subjective segmentation of diffuse gliomas. Future research should expand to include subregions such as necrosis zones, peritumoral zones, and anatomical structures surrounding the tumor. Our long-term goal is to implement these models in real-time clinical settings, supporting not only pre-resection segmentation but also intraoperative imaging to detect and segment residual tumor tissue—a development with high potential impact in neuro-oncological surgery.

## 5. Conclusions

This study presents a multicenter-trained model for glioma segmentation on intraoperative ultrasound (ioUS) images, achieving robust generalizability and reproducibility across diverse external datasets, even those with lower image quality. Key contributions include the use of a high-quality multicenter dataset with carefully curated segmentation annotations, a robust framework for model training, and external validation to ensure reproducibility. These strengths highlight the potential of our approach for addressing the challenges of glioma segmentation in neuro-oncological surgery.

Future improvements will focus on expanding the dataset to include a larger and more diverse cohort, incorporating segmentation of tumor subregions, and developing real-time capabilities for intraoperative use. These advancements aim to further enhance the role of ioUS in glioma resection, ultimately improving surgical outcomes for patients.

## Figures and Tables

**Figure 1 cancers-17-00315-f001:**
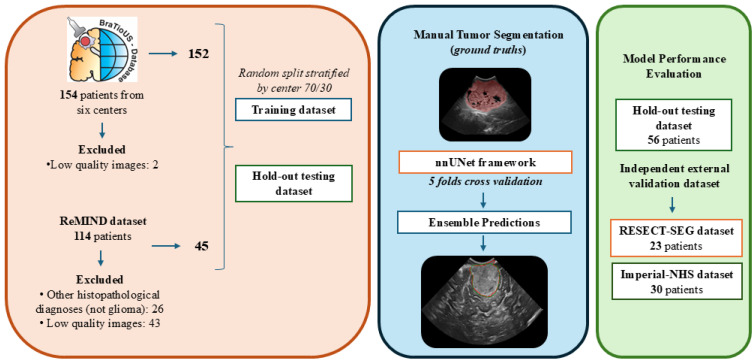
Schematic representation of the workflow followed in this study.

**Figure 2 cancers-17-00315-f002:**
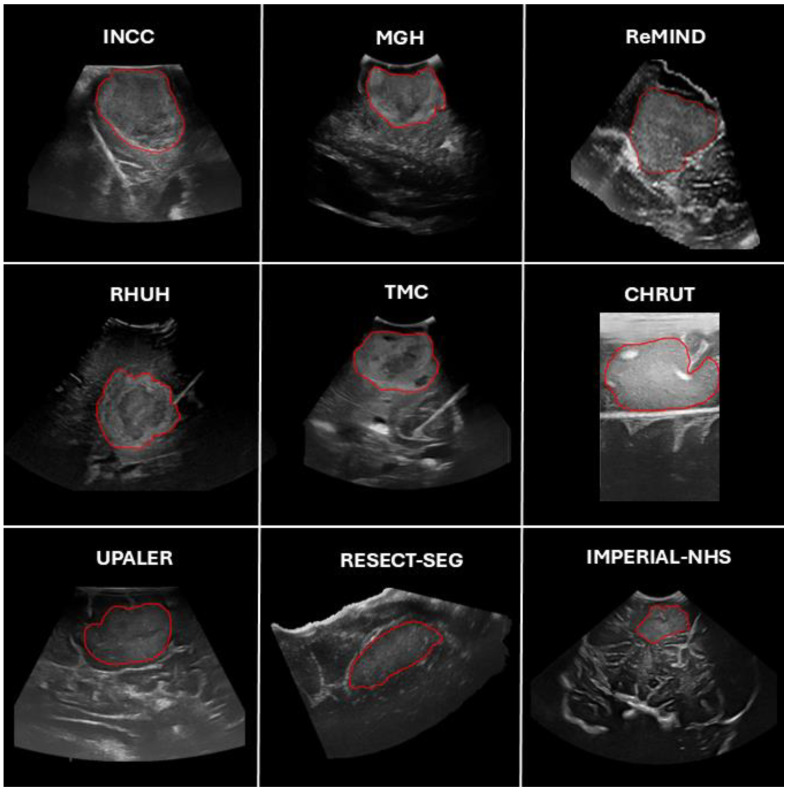
Representative examples of patients from the different datasets and centers included in the study. Tumor segmentations, considered the ground truth, are highlighted with red contours.

**Figure 3 cancers-17-00315-f003:**
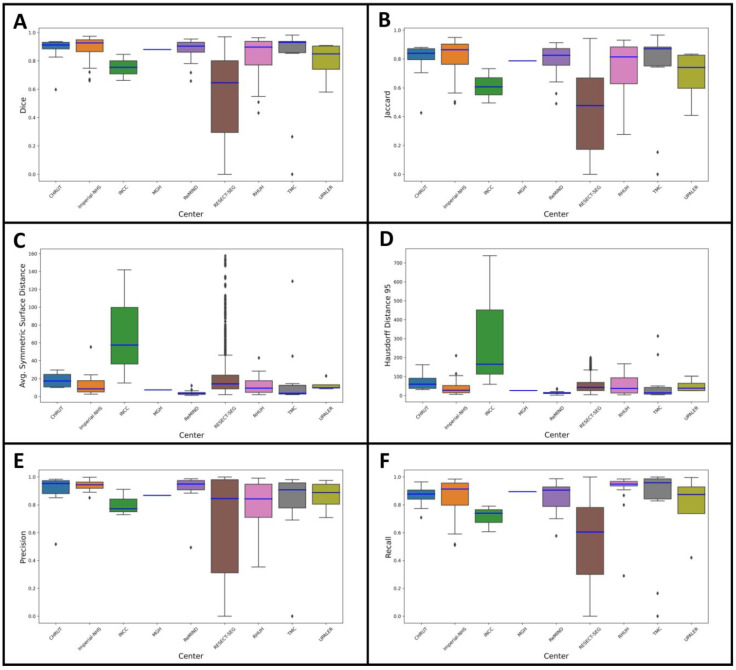
Performance metrics of the glioma segmentation model across different centers. Each subplot represents a key evaluation metric: (**A**) Dice similarity coefficient, (**B**) Jaccard index, (**C**) average symmetric surface distance, (**D**) 95th percentile Hausdorff distance, (**E**) precision, and (**F**) recall. Boxplots illustrate the distribution of metric scores for each center, with the blue horizontal line indicating the median value for each center. Outliers are represented as individual points outside the whiskers.

**Figure 4 cancers-17-00315-f004:**
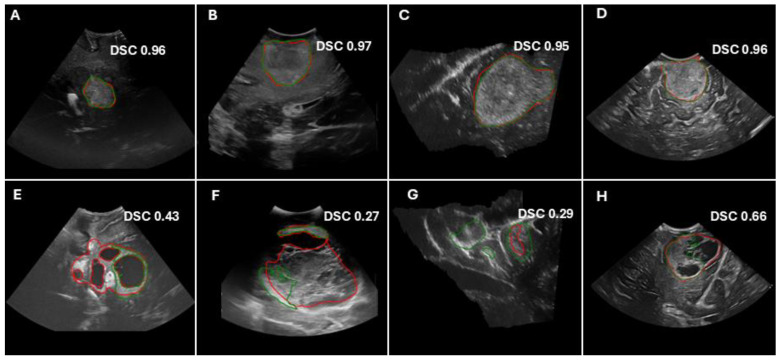
Examples of model predictions and Dice similarity score (DSC) values for the hold-out test cohorts (**A**,**B**,**E**,**F**), as well as the external validation cohorts (**C**,**G**) (RESECT-SEG) and (**D**,**H**) (Imperial-NHS). The top panels show cases with good performance, whereas the bottom panels illustrate cases with poor performance. The ground truth tumor segmentations are delineated in red contours, whereas predicted segmentations are shown in green.

**Table 1 cancers-17-00315-t001:** Demographic characteristics and ultrasound image acquisition details across centers.

Variable	Datasets/Centers						
	RHUH	ReMIND	TMC	CHRUT	UPALER	INCC	MGH
Total of subjects	58	45	35	29	15	10	5
Mean Age	61.66 ± 11.22	42.49 ± 15.16	47.62 ± 10.72	48.16 ± 13.76	67.33 ± 12.98	58.9 ± 18.77	NA
Sex							NA
Male	36 (62.07%)	17 (37.78%)	24 (68.57%)	9 (30%)	5 (33.33%)	6 (60%)	
Female	22 (37.93%)	28 (62.22%)	11 (31.43%)	13 (43.33%)	10 (66.67%)	4 (40%)	
NA	-	-	-	8 (26.67%)	-	-	
WHO grade							
2	-	11 (24.44%)	-	8 (27.59%)	-	-	-
3	-	12 (26.67%)	-	8 (27.59%)	-	-	-
4	58 (100%)	16 (35.56%)	35 (100%)	13 (44.83%)	10 (66.67%)	10 (100%)	5 (100%)
NA	-	6 (13.33%)	-	-	5 (33.33%)	-	NA
IDH status							
Mutant	8 (13.79%)	23 (51.1%)	4 (11.43%)	9 (31.03%)	-	-	-
Wild-Type	50 (86.21%)	16 (35.56%)	31 (88.57%)	11 (44.83%)	-	10 (100%)	-
NA	-	6 (13.33%)	-	7 (24.14%)	15 (100%)	-	5 (100%)
US manufacturer	Hitachi	BK	BK/Sonowand AS	Supersonic	Esaote	Esaote	BK
Type of probe	Curved	Curved	Curved	Linear	Linear	Linear	Curved
Frequency	4–8 Mhz	5–13 Mhz	3–8 Mhz	4–15 Mhz	3–11 Mhz	3–11 Mhz	5–13 Mhz
Acquisition type							
2D	58 (100%)	-	11 (31.42%)	29 (100%)	15 (100%)	10 (100%)	5 (100%)
3D	-	45 (100%)	24 (68.57%)	-	-	-	-

The values are expressed as standard deviation and percentages as applicable. WHO = World Health Organization. IDH = isocitrate dehydrogenase. US = ultrasound. MHz = megahertz.

**Table 2 cancers-17-00315-t002:** Performance evaluation across datasets and centers.

Hold-Out Testing Cohort							
Center/Dataset	Number of Patients	ASSD	DSC	HD 95	IoU	Precision	Sensitivity
All	56	8.51 ± 1.63	0.9 ± 0.01	29.08 ± 7.02	0.82 ± 0.02	0.91 ± 0.02	0.91 ± 0.02
RHUH	17	9.48 ± 3.7	0.9 ± 0.04	38.36 ± 17.89	0.81 ± 0.06	0.84 ± 0.06	0.95 ± 0.01
ReMIND	13	3.61 ± 0.64	0.9 ± 0.03	13.0 ± 2.19	0.83 ± 0.04	0.95 ± 0.02	0.91 ± 0.04
TMC	10	3.68 ± 5.91	0.93 ± 0.09	14.52 ± 27.41	0.87 ± 0.1	0.91 ± 0.05	0.96 ± 0.11
CHRUT	8	17.38 ± 4.91	0.91 ± 0.03	60.72 ± 19.54	0.84 ± 0.04	0.95 ± 0.04	0.88 ± 0.03
UPALER	4	9.85 ± 4.0	0.85 ± 0.09	39.65 ± 22.33	0.74 ± 0.12	0.89 ± 0.07	0.87 ± 0.15
INCC	3	57.53 ± 46.45	0.76 ± 0.07	164.83 ± 266.57	0.61 ± 0.09	0.77 ± 0.07	0.74 ± 0.07
MGH	1	7.46 ± 0.0	0.88 ± 0.0	27.22 ± 0.0	0.79 ± 0.0	0.87 ± 0.0	0.89 ± 0.0
**External validation cohorts**							
RESECT-SEG	23	14.14 ± 0.23	0.65 ± 0.01	44.02 ± 0.76	0.48 ± 0.01	0.84 ± 0.01	0.61 ± 0.01
Imperial-NHS	30	8.58 ± 1.78	0.93 ± 0.01	28.8 ± 7.62	0.86 ± 0.02	0.94 ± 0.01	0.91 ± 0.03

ASSD = average symmetric surface distance. DSC = dice similarity coefficient. HD 95 = Hausdorff distance 95th percentile. IoU = intersection over union. Values are expressed as median ± 95% Confidence Interval (bootstrapped).

## Data Availability

The data presented in this study are available on request from the corresponding author due to patient privacy and ethical restrictions.

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
