# Peer review of "Deep Learning-Based Glioma Segmentation of 2D Intraoperative Ultrasound Images: A Multicenter Study Using the Brain Tumor Intraoperative Ultrasound Database (BraTioUS)"

_cancers, 2025, doi:10.3390/cancers17020315_

Round 1

Reviewer 1 Report

Comments and Suggestions for Authors

Thank the authors for the submission of the manuscript. The paper is well-organized, written in a fluent language and falls within the scope of the journal. Basic contribution of presented study lies in multicenter data collection for glioma segmentation on ioUS images since algorihmic scheme is based on an already established CNN architecture. Presentation of related work is compact and based on recent references, revealing limitations and challenges with respect to the examined scientifi field. Quantitative results provided appear promising.

In order to further improve paper quality, the following amendments are proposed:

* please enrich description of paper contribution at the end of the Inroduction Section

* please add technical details on the development of the nnU-Net framework so that reproducibility of results is feasible by any interested researcher. Towards this, you may answer the following key questions/issues:

   - own contribution in the implementation of the cnn?

  - illustration of overall architecture in an informative figure, indicating discrete components and setup

   - features for training (e.g. raw images, dimensions, any pre-processing)?

   - mathematical formula for loss function

   - size of dataset after data augmentation

*  significance of proposed work could be enhanced making data available under open access license upon applying anonymity protocols. You may consider this approach

* please include comparative analysis (qualitatively and/or quantitatively) of proposed work against similar approaches towards glioma segmentation on ioUS images

* please further enhance the Conclusions Section summarizing key contribution of presented work, "take-home messages" and specific details on future improvement

* please add a short description at the end of the Conclusions Section on future improvement of presented study 

* please use a uniform formatting for references. Also, please provide full citation information for references [7]. [13], [22], [29]

Author Response

Comments 1: please enrich description of paper contribution at the end of the Introduction Section

Response 1: We appreciate the reviewer’s suggestion. To address this, we have expanded the final paragraph to explicitly outline the novel contributions and clinical relevance of our work, emphasizing its significance in advancing glioma segmentation within the intraoperative ultrasound (ioUS) domain. The revised text is as follows:

 “In this study, we aim to address the challenge of accurate and automated glioma segmentation in ioUS images by developing a CNN model trained on a multicenter dataset provided by the Brain Tumor Intraoperative Ultrasound Database (BraTioUS). This dataset includes images acquired using various scanners and diverse acquisition protocols, enabling the model to handle the inherent variability of ioUS data. Our work focuses on both high- and low-grade gliomas, with the aim of evaluating the model's performance on external datasets to ensure its reproducibility and robustness. This approach seeks to overcome key limitations in existing methodologies, advancing the interpretability and clinical applicability of ioUS for glioma surgery”

Comments 2: please add technical details on the development of the nnU-Net framework so that reproducibility of results is feasible by any interested researcher. Towards this, you may answer the following key questions/issues:

a) own contribution in the implementation of the cnn?

Response 2a: We appreciate the reviewer’s emphasis on ensuring reproducibility and transparency. Regarding the implementation of the convolutional neural network (CNN), we would like to clarify that we did not modify the architecture or the underlying framework settings. All configurations, including preprocessing, data augmentation, training hyperparameters, and postprocessing, were applied using the default settings provided by nnU-Net. This choice was intentional, as one of the strengths of nnU-Net lies in its ability to automatically adapt its configurations to the dataset at hand, offering a robust and reproducible baseline for medical image segmentation tasks. By adhering strictly to the default settings, we aimed to minimize biases introduced by manual optimization and to ensure that our results can be easily reproduced by other researchers following the same approach.

b) illustration of overall architecture in an informative figure, indicating discrete components and setup

Response 2b: The CNN used in our study is based on the widely recognized U-Net architecture, which has been extensively documented in the literature (Ronneberger et al.). Since we adhered to the original U-Net architecture without modifications, we believe that adding a figure illustrating this architecture would be redundant, as it is already well-known to the intended audience. Moreover, we aimed to maintain a balance between technical details and the clinical focus of this journal. Including such a figure might shift the emphasis away from the practical and clinical implications of our work, which we consider central to the manuscript's contribution. Additionally, the technical details regarding the U-Net architecture are thoroughly described in the references already cited in the manuscript, ensuring that interested readers can access the necessary information.

We trust that this approach aligns with the expectations of the journal’s readership, but we remain open to further suggestions from the reviewers.

c) features for training (e.g. raw images, dimensions, any pre-processing)?    

Response 2c: We thank the reviewer for requesting further clarification on the features used during training. Our study utilized the default configurations of the nnU-Net framework without modifications. The dataset, consisted of 2D ultrasound images with a median input dimension of 600 × 800 pixels after cropping, and the images were processed using the SimpleITKIO library. Preprocessing steps included Z-Score normalization, where the foreground intensity was standardized with a mean of 93.39 and a standard deviation of 38.78, ensuring a mean of zero and unit variance. Images were also resampled to a consistent spacing of 1.0 mm × 1.0 mm, and cropped to retain regions of interest, resulting in a median shape of 600 × 800 pixels.

For training, nnU-Net automatically configured the patch size as 640 × 896 pixels with a batch size of six. The data augmentation pipeline included random rotations, scaling, flipping, and elastic deformations. The underlying CNN architecture used was a U-Net, with eight stages and features per stage increasing from 32 to 512. Each layer employed a 3 × 3 convolutional kernel and downsampling strides of 2 in both dimensions. The network utilized instance normalization with eps=1e-05 and affine parameters, along with Leaky ReLU as the activation function. Batch Dice loss was enabled during training to optimize segmentation performance.

Foreground intensity properties of the images were also analyzed, with minimum and maximum intensities of 0.0 and 255.0, respectively, a median intensity of 87.0, and a 0.5–99.5th percentile intensity range of 18.0–206.0. These configurations and preprocessing steps were determined and applied automatically by nnU-Net, ensuring reproducibility. Since no modifications were made to these defaults, the details provided here, along with references to the nnU-Net framework, should enable any interested researcher to replicate our results.

d) mathematical formula for loss function

Response 2d: We appreciate the reviewer's suggestion to include the mathematical formulas for the loss functions employed in our study. In response, we have expanded the Methods section to provide these details. Specifically, we utilized the default nnU-Net configuration, which combines the Soft Dice Loss and Cross-Entropy Loss to optimize segmentation performance.

e) size of dataset after data augmentation

Response 2e: We appreciate the reviewer's inquiry regarding the dataset size after data augmentation. In our study, we employed the nnU-Net framework, which applies data augmentation techniques in real-time during the training process. This on-the-fly augmentation means that each training epoch is exposed to a unique, augmented version of the original dataset. As a result, the augmented samples are not explicitly stored but are generated in real time. Consequently, while the number of original training samples remains constant, the diversity and variability introduced through augmentation significantly enhance the model's robustness and generalization capabilities. This approach ensures that the model learns to perform well across a wide range of potential variations without the need to explicitly store additional augmented data, thereby maintaining computational efficiency. Because of this approach, the exact size of the dataset after augmentation cannot be determined. For a comprehensive understanding of nnU-Net's data augmentation strategies and their implementation, we refer to the nnU-Net repository and associated documentation https://github.com/MIC-DKFZ/nnUNet

f) significance of proposed work could be enhanced making data available under open access license upon applying anonymity protocols. You may consider this approach

Response 2f: We acknowledge the reviewer's valuable suggestion. However, the BraTioUS dataset used in our study is a multicenter dataset collected under ethical guidelines specific to each contributing institution. Currently, these guidelines do not permit unrestricted public access to the data. We recognize the importance of open access for advancing research and are committed to working towards making the dataset publicly available in the medium term. This process will require obtaining approval from each contributing center, as well as significant administrative efforts to navigate the requirements of their respective Institutional Review Boards (IRBs). While this will take a considerable time, we aim to ensure compliance with all ethical and regulatory standards as we pursue this goal.

We trust this explanation provides clarity regarding the current limitations and our future intentions.

Comments 3: please include comparative analysis (qualitatively and/or quantitatively) of proposed work against similar approaches towards glioma segmentation on ioUS images

Response 3: We appreciate the reviewer's suggestion. However, to the best of our knowledge, the number of published methods addressing tumor segmentation in ioUS is very limited, and the corresponding models and implementations are not publicly available, making direct comparison challenging.

Moreover, a key dataset used in these few published works is the RESECT-SEG dataset, which has primarily been employed for training and validation purposes in those studies. In contrast, we utilized the RESECT-SEG dataset exclusively for external testing in our work to evaluate the generalizability and robustness of our model. As such, even if these models were publicly available, a direct comparison would be methodologically inconsistent due to the differing roles of the dataset in the respective workflows.

Comments 4: a) please further enhance the Conclusions Section summarizing key contribution of presented work, "take-home messages" and specific details on future improvement. b) please add a short description at the end of the Conclusions Section on future improvement of presented study 

Response 4 a-b: Thank you for the constructive feedback on improving the Conclusions section. We have updated it to emphasize the key contributions, take-home messages, and future directions of our work.

Conclusions

This study presents a multicenter-trained model for glioma segmentation on intraoperative ultrasound (ioUS) images, achieving robust generalizability and reproducibility across diverse external datasets, even those with lower image quality. Key contributions include the use of a high-quality multicenter dataset with carefully curated segmentation annotations, a robust framework for model training, and external validation to ensure reproducibility. These strengths highlight the potential of our approach for addressing the challenges of glioma segmentation in neuro-oncological surgery.

Future improvements will focus on expanding the dataset to include a larger and more diverse cohort, incorporating segmentation of subregions, and developing real-time capabilities for intraoperative use. These advancements aim to further enhance the role of ioUS in glioma resection, ultimately improving surgical outcomes for patients.”

Comments 5: please use a uniform formatting for references. Also, please provide full citation information for references [7]. [13], [22], [29]

Response 5: Thank you for your feedback. The formatting has been carefully reviewed and updated to align with the MDPI journal requirements. Additionally, the full citation details have been included in the revised manuscript.

Reviewer 2 Report

Comments and Suggestions for Authors

The following comments are to be incorporated in the revised manuscript:

1. The author has to define the problem statement in the introduction section and include the manuscript structure.

2. The author has to create a separate literature review section that contains the strengths and limitations of the previously published work.

3. The collected dataset is annotated by an expert medical expert.  It is suggested that some other experts must verify labeled and marked images. So that the ground truth region is more accurate and precise.

4. In this manuscript, the author used nnU-Net model for segmentation. The author has to explain that the considered nnU-Net model differs from the basic U-Net model.

5. The author has to explain the layer structure of the modified nnU-Net.

6. The author has to add more Evaluation Metrics. (Author can get help from DOI: 10.3233/JIFS-189773)

7. In the result section, there should be a comparative analysis between previous work and the current study.

8. In the methodology section, there should be an experimental workflow diagram for better understanding and improving readability.

9. There should be language proofing required.

10. All acronyms must be defined at their first appearance.

11. The author has to highlight their contributions.

12. In the domain of malignancy segmentation, a healthy amount of studies are available. How the presented work is unique and different from the existing one?

Comments on the Quality of English Language

Minor corrections required.

Author Response

Comments 1: The author has to define the problem statement in the introduction section and include the manuscript structure.

Response 1: We thank the reviewer for their valuable suggestion to define the problem statement in the introduction. In response, we have revised the introduction to explicitly define the specific challenges of intraoperative ultrasound (ioUS) in neurosurgery, including its recognized limitations in image quality, interpretability, and operator dependence. These challenges highlight the need for solutions like automated tumor segmentation that enhance the usability of ioUS without compromising its affordability and simplicity. This revision aims to provide clear context and motivation for our study.

Comments 2: The author has to create a separate literature review section that contains the strengths and limitations of the previously published work.

Response 2: We sincerely thank the reviewer for their valuable comment regarding the inclusion of a dedicated literature review section to discuss the strengths and limitations of previously published work. To maintain the concise structure of the manuscript and avoid excessive length, we have decided not to create a separate subsection. However, we have incorporated the required information into the introduction, ensuring that the strengths and limitations of prior studies are thoroughly addressed and that the context and motivation for our work are clearly presented.

We trust this approach aligns with the reviewer’s expectations while maintaining the overall balance and flow of the manuscript.

Comments 3: The collected dataset is annotated by an expert medical expert.  It is suggested that some other experts must verify labeled and marked images. So that the ground truth region is more accurate and precise.

Response 3: Thank you for highlighting this important point. It is indeed challenging to eliminate the risk of bias in studies involving manual segmentations by annotators. In a previous study (PMID: 33594589), we assessed inter-observer agreement in our tumor segmentations on intraoperative ultrasound (ioUS) and found a high level of similarity, with a Dice score of 0.9. Based on this prior evidence, in the current study, the responsibility for the annotations was entrusted to the first author, whose unique profile combines expertise in image processing, segmentation, and the intraoperative application of this imaging modality in daily clinical practice.

Recognizing that bias is inevitable whether performed by one or multiple observers, our team opted for a single annotator for this study, considering that the data volume was manageable and that this approach ensured a methodical and consistent quality in the annotations.

Comments 4: In this manuscript, the author used nnU-Net model for segmentation. The author has to explain that the considered nnU-Net model differs from the basic U-Net model.

Response 4: We thank the reviewer for their observation regarding the need to explain how the nnU-Net framework differs from the basic U-Net model. We would like to clarify that nnU-Net is not a model or a specific architecture but rather a self-configuring framework designed for medical image segmentation tasks. As described in the "nnU-Net Framework" section of the manuscript, nnU-Net uses the U-Net architecture as its foundation but goes beyond it by automating key aspects such as preprocessing, network configuration, and hyperparameter optimization.

This automated framework allows nnU-Net to adapt its configurations to the specific characteristics of a dataset and task, in contrast to the basic U-Net model, which requires manual design and tuning. These capabilities are crucial for ensuring the model's effectiveness and robustness across diverse datasets, as demonstrated in this study.

We trust this explanation addresses the reviewer’s concern and provides clarity regarding the role of nnU-Net in our work.

Comments 5: The author has to explain the layer structure of the modified nnU-Net.

Response 5: We value the reviewer’s detailed observation on this matter. As mentioned in the manuscript, we used the nnU-Net framework in its default 2D configuration without modifying the layer structure. Detailed descriptions of the U-Net architecture and the nnU-Net framework are available in the referenced works [22, 23]. We trust this addresses the reviewer’s concern.

Comments 6: The author has to add more Evaluation Metrics. (Author can get help from DOI: 10.3233/JIFS-189773)

Response 6: We appreciate the reviewer’s comment regarding the evaluation metrics used in this study. To comprehensively assess the model's performance, we utilized the USE-Evaluator, which provides advanced overlap and distance metrics, such as Dice similarity coefficient (DSC), intersection over union (IoU), 95th percentile Hausdorff distance (HD 95), and average symmetric surface distance (ASSD). Additionally, we included precision and sensitivity to evaluate the model’s ability to correctly identify positive and negative cases. We believe this combination of metrics provides a comprehensive and robust evaluation of segmentation performance, effectively capturing both spatial accuracy and boundary delineation precision. Given that no specific metrics were requested, we consider this approach sufficient for assessing the model’s performance.

Comments 7: In the result section, there should be a comparative analysis between previous work and the current study.

Response 7: We acknowledge the reviewer’s insightful remark. While we acknowledge the value of such comparisons, we believe it is not feasible to perform a formal benchmarking analysis due to the significant disparities in datasets, CNN architectures, and experimental setups across studies. These differences make it challenging to establish a fair and standardized comparison.

Instead, we have discussed the results of our study in the Discussion section, where we highlight the contributions and contextualize our findings in relation to previous work. This approach ensures that the nuances of our dataset and methodology are adequately considered. We trust this explanation addresses the reviewer’s concern.

Comments 8: In the methodology section, there should be an experimental workflow diagram for better understanding and improving readability.

Response 8: We thank the reviewer for bringing up this important point. We would like to point out that the current manuscript already includes a figure that schematically represents the experimental workflow in a clear and concise manner. Figure 1 was designed to provide an intuitive understanding of the methodology without adding unnecessary complexity. We trust that this figure effectively communicates the workflow, but we are happy to make adjustments if the reviewer feels additional details are necessary.

Comments 9: There should be language proofing required.

Response 9: We are grateful for the reviewer’s valuable feedback. We would like to clarify that the entire manuscript has been reviewed by several of our co-authors, who are native English speakers. Additionally, the paper has undergone a grammar check by a professional language editing service to ensure clarity and correctness. We trust that these steps have addressed any potential language issues.

Comments 10: All acronyms must be defined at their first appearance.

Response 10: We appreciate the reviewer’s thoughtful observation. We have carefully reviewed the manuscript to ensure that all acronyms are defined at their first appearance. If there are any specific instances that were overlooked, we would be happy to address them.

Comments 11: The author has to highlight their contributions.

Response 11: We thank the reviewer for their suggestion to highlight the contributions of our work. We would like to point out that these contributions are already emphasized in the Discussion section, where we contextualize our findings and their relevance within the field.

We have intentionally opted not to expand further to maintain a balanced tone and avoid overstatements. We believe this approach ensures that the manuscript remains objective and focused while effectively communicating the significance of our work.

Comments 12: In the domain of malignancy segmentation, a healthy amount of studies are available. How the presented work is unique and different from the existing one?

Response 12: We thank the reviewer for their insightful comment. It is true that a considerable amount of literature exists on tumor segmentation. However, as mentioned in the Introduction, most of these studies focus on MRI-based segmentation. There are only seven publications dedicated to ultrasound segmentation of brain tumors, and of these, only three specifically focus on gliomas.

As highlighted in both the Introduction and Discussion sections, our work is unique in several ways: it is the first to use a multicenter dataset, focuses on high-quality native 2D ultrasound images, includes gliomas of both high and low grade, and performs external cohort validation to ensure robustness and reproducibility. These aspects set our study apart from existing works in the field.

Reviewer 3 Report

Comments and Suggestions for Authors

The authors present a deep learning model for glioma segmentation on intraoperative ultrasound (ioUS) images, leveraging multicenter data to validate the approach. While the manuscript is well-written and addresses an important clinical need, several areas require further clarification, additional experiments, and structural improvements before it can be considered for publication. 

1. Include a subgroup analysis that evaluates the segmentation performance (e.g., DSC, ASSD) by center or acquisition protocol. A table correlating imaging parameters with performance metrics would strengthen the study.

2. The RESECT-SEG dataset shows significantly lower segmentation performance compared to the Imperial-NHS cohort (DSC: 0.65 vs. 0.93). This discrepancy is mentioned but not analyzed in sufficient detail. Provide a detailed discussion of potential reasons (e.g., imaging quality, glioma grade distribution, annotation methods). Include representative visualizations of poor segmentations to highlight failure modes.

Minor comments:

1. The selection of the "largest tumor slice" is mentioned but not justified. Explain why this slice was chosen and whether this approach could introduce bias in representing tumor heterogeneity.

2. Minor suggestion, add a plot or table summarizing performance metrics (e.g., DSC, ASSD) for each center to provide a clearer picture of variability. 

Author Response

Comments 1: Include a subgroup analysis that evaluates the segmentation performance (e.g., DSC, ASSD) by center or acquisition protocol. A table correlating imaging parameters with performance metrics would strengthen the study.

Response 1: We thank the reviewer for this valuable suggestion. We would like to point out that the results of the model’s performance evaluation grouped by center are already presented in Table 2. Additionally, the acquisition protocol for each center is detailed in Table 1. We believe these tables provide the information requested and effectively correlate imaging parameters with performance metrics, as suggested.

Comments 2: The RESECT-SEG dataset shows significantly lower segmentation performance compared to the Imperial-NHS cohort (DSC: 0.65 vs. 0.93). This discrepancy is mentioned but not analyzed in sufficient detail. Provide a detailed discussion of potential reasons (e.g., imaging quality, glioma grade distribution, annotation methods). Include representative visualizations of poor segmentations to highlight failure modes.

Response 2: We thank the reviewer for their insightful comment. As mentioned in the manuscript, we have already explained the potential reasons for the lower performance in the RESECT-SEG dataset, including the variability in ground truth segmentations, lower spatial and temporal resolution of the 3D volumes, and the predominance of low-grade gliomas with diffuse boundaries, which are inherently more challenging to delineate.

While we have been respectful of the authors of the RESECT-SEG dataset, it is important to acknowledge that the image quality is questionable compared to the Imperial-NHS cohort. Nevertheless, we deemed it valuable to include this external dataset in our study, given its prior use by other authors and its relevance as an independent validation source. The most straightforward approach would have been to exclude it and rely solely on the higher-quality Imperial-NHS cohort; however, we believe it is more transparent and scientifically rigorous to present all results, even when the performance is suboptimal.

Comments 3: The selection of the "largest tumor slice" is mentioned but not justified. Explain why this slice was chosen and whether this approach could introduce bias in representing tumor heterogeneity.

Response 3: We appreciate the reviewer’s thoughtful comment regarding the selection of the "largest tumor slice" and its potential implications. The decision to use only one image per patient was based on the following considerations: This is a retrospective study involving multiple centers and a highly operator-dependent imaging modality, making it very challenging to establish a common acquisition pattern for the ultrasound images.

Ideally, we would have preferred to include all available images per subject to better capture tumor heterogeneity. However, in most cases, we observed a mixture of images where the tumor was only partially captured, and orthogonal planes had not been consistently acquired, which would have been optimal. To address this inconsistency, we opted to focus on the largest tumor slice for each patient, following the methodology established in one of our previous studies (PMID: 33594589).

We acknowledge the limitations of this approach and recognize that it may not fully capture the heterogeneity of the tumors. However, this was a pragmatic choice to ensure consistency across the dataset and comparability of results.

Comments 4: Minor suggestion, add a plot or table summarizing performance metrics (e.g., DSC, ASSD) for each center to provide a clearer picture of variability. 

Response 4: We appreciate the reviewer’s valuable suggestion. In response, we have added a figure (Figure 3) summarizing the performance metrics (e.g., Dice Similarity Coefficient, ASSD) for each center. This figure provides a clear visualization of the variability across centers and complements the detailed results already presented in the manuscript. We believe this addition enhances the clarity and presentation of our findings.

Round 2

Reviewer 1 Report

Comments and Suggestions for Authors

Thank the authors for the submission of the revised manuscript. Amendments proposed under the initial review procedures have been addressed adequately and key information, where considered appropriate, has been incorporated into the corresponding section. Open access of dataset, due to time-intensive procedures required, has been decided as a future step towards enhancing significance of presented study, while technical details on the development of the deep neural network for image analysis are totally available via referenced works. I welcome the manuscript for publication in the journal and I wish the authors all the best to their future plans.